# A Novel Variant in RAD21 in Cornelia De Lange Syndrome Type 4: Case Report and Bioinformatic Analysis

**DOI:** 10.3390/genes14010119

**Published:** 2023-01-01

**Authors:** Alessandro De Falco, Daniele De Brasi, Matteo Della Monica, Claudia Cesario, Stefano Petrocchi, Antonio Novelli, Giuseppe D’Alterio, Achille Iolascon, Mario Capasso, Carmelo Piscopo

**Affiliations:** 1U.O.C. Genetica Medica, A.O.U. Federico II, 80131 Naples, Italy; 2Department of Molecular Medicine and Medical Biotechnology, University Federico II, 80131 Naples, Italy; 3U.O.C. Pediatria, AORN Santobono Pausilipon, 80129 Naples, Italy; 4Medical and Laboratory Genetic Unit, Antonio Cardarelli Hospital, 80131 Naples, Italy; 5Laboratory of Medical Genetics, Bambino Gesù Children Hospital IRCCS, 00165 Rome, Italy; 6European School of Molecular Medicine, Università Degli Studi di Milano, 20139 Milan, Italy; 7CEINGE Biotecnologie Avanzate, 80145 Naples, Italy

**Keywords:** Cornelia de Lange type 4, *RAD21*, *in silico* analysis, genotype/phenotype correlation, prenatal growth retardation

## Abstract

Cornelia de Lange Syndrome (CdLS) is a rare genetic disorder that affects many organs. The diagnosis of this condition is primarily clinical and it can be confirmed by molecular analysis of the genes known to cause this disease, although about 30% of CdLS patients are without a genetic diagnosis. Here we report clinical and genetic findings of a patient with CdLS type 4, a syndrome of which the clinical features of only 30 patients have been previously described in the literature. The index patient presented with clinical characteristics previously associated with CdLS type 4 (short nose, thick eyebrow, global development delay, synophrys, microcephaly, weight < 2DS, small hands, height < 2DS). She also presented cardiac anomalies, cleft palate and laryngomalacia, which was never described before. The index patient was diagnosed with a novel *de novo RAD21* variant (c.1722_1723delTG, p.Gly575SerfsTer2): segregation analysis, bioinformatic analysis, population data and in silico structural modelling indicate the pathogenicity of the novel variant. This report summarizes previously reported clinical manifestations of CdLS type 4 but also highlights new clinical symptoms, which will aid correct counselling of future CdLS type 4 cases.

## 1. Introduction

Cornelia de Lange Syndrome (CdLS; MIM #122470, 300590, 610759, 614701, 300882) is a rare genetic disorder that affects many organs. This multi-malformative condition is characterized by pre- and postnatal growth retardation, psychomotor and intellectual disability (mild to severe), distinctive dysmorphic features, hirsutism and malformations of especially the upper limbs [1].

The diagnosis of this condition is primarily clinical: a diagnostic algorithm for CdLS was published in 2018 [2]. Major (or cardinal) and minor (or suggestive) criteria have been established, giving a score of two points and one point, respectively, if the patient presents them; depending on the sum of the clinical score obtained, a spectrum of phenotypes of CdLS can be identified (Table 1).

Confirmation of the clinical diagnosis can be obtained by molecular analysis of the genes known to cause this condition, although about 30% of CdLS patients are without a genetic diagnosis. The CdLS rainbow spectrum includes patients with the classic and non-classic CdLS phenotype who have a pathogenic variant in a gene involved in cohesin functioning. There are other patients who share limited signs of CdL phenotype and there is a presumed pathogenic variant in a cohesion function-relevant gene that is unknown at the moment (Figure 1). The identified mutations are related to genes that code for proteins belonging to the cohesin complex [3]: the most mutated gene is *NIPBL* [4], followed by *SMC1A* [5]. In recent years, pathogenic variants have been described in the genes belonging to the cohesin complexes *SMC3* [6], *HDAC8* [7], *RAD21* [8] and currently it is not possible to establish the real prevalence of mutations of these three genes in patients affected by this condition due to few clinical and molecular descriptions of these patients.

There are also patients affected by non-classic CdLS with pathogenetic variants in *BRD4* and *ANKRD11* [9] genes.

The cohesin complex is involved in the segregation of sister chromatids during cell division [10]. The cohesion ring, a multi-protein structure, is composed of Smc1A, Smc3, Rad21 and a STAG subunit and holds the sister chromatids together along their length and at centromere until segregation occurs [11]. Impairment of the cohesin complex is associated with cohesinopathies in humans, including Roberts Syndrome and CdLS. In total, around 500 mutations affecting the cohesin complex have been associated with CdLS [1]. However, only few different alterations in *RAD21* have been identified as giving rise to CdLS, referred to as CdLS type 4 (MIM #614701).

*RAD21* was first associated with CdLS type 4 in four unrelated CdLS patients [12]. Recently, a large study of correlation between genotypes and phenotypes of patients affected by CdLS type 4 and variants in the *RAD21* gene has been conducted [13]: full clinical information is available for 30 individuals. Recently, a case of holoprosencephaly associated with the RAD21 loss-of-function variant was described [14]. Other authors reported holoprosencephaly associated with variants in *STAG2*, *SMC1A*, *SMC3* and *RAD21*. These findings indicate the cohesin complex is an important regulator of median forebrain development [15].

Herein, we expand the clinical description of CdLS type 4 by reporting the clinical features of a 3-year-old girl with a novel mutation in *RAD21* and comparing our case with those previously reported. We also highlight the effect of the variant by *in silico* structural modelling.

## 2. Materials and Methods

Ethical review and approval were waived for this study, dealing with a case report conducted according to clinical practice guidelines. Written informed consent has been approved from the patient(s) to publish this paper. Genomic DNA of proband and her parents were isolated from peripheral blood using the QIAsymphony DSP DNA Mini Kit (Qiagen, Hilden, Germany) following manufacturer’s instructions. Library preparation and clinical exome capture were performed *in trio* using the Twist Custom Panel kit (Twist Bioscience, San Francisco, CA, USA) and sequenced on the NovaSeq 6000 platform (Illumina, San Diego, CA, USA). The BaseSpace pipeline (Illumina) and the Geneyx Analysis (Knowledge-Driven NGS Analysis tool powered by the GeneCards Suite) were used for variant calling and annotation. Reads were aligned to human genome build GRCh37/hg19. Based on the guidelines of the American College of Medical Genetics and Genomics and the Association for Molecular Pathology (ACMG/AMP), a minimum depth coverage of 20X was considered suitable for analysis. Subsequent variant annotation was performed with the Variant Effect Predictor (VEP) web tool [16]. Technical validation of the results was performed by Sanger analysis with BigDye^®^ Terminator v3.1 kit, capillary electrophoresis and the automatic sequencing platform Genetic Analyzer (Applied Biosystems, Waltham, MA, USA). Three-dimensional conformation of wild type and mutant Rad21 was retrieved by Protein Data Bank (PDB) database [17] and the Phyre2 [18] web tool, respectively, and visualized with PyMOL v2.5.

## 3. Results

### 3.1. Proband Phenotype

The patient was the second child of healthy non-related parents (Figure 2). The first child was in good health and had no dysmorphic features, no intellectual disability, no growth retardation or other health problems. The patient was born at 36 weeks of gestational age by urgent caesarean section due to maternal gestosis. The birthweight was 1980 g (Small for Gestational Age, SGA; second–ninth percentile), the length was 43 cm (second–ninth percentile), the head circumference 27.2 cm (-2DS) and the Apgar Score was 2-5-7. She needed oxygen therapy and endo-tracheal intubation.

She was admitted to neonatal intensive care for 6 months and the following problems emerged: cardiac anomalies (patent ductus arteriosus, PDA, and ventricular septal defect, VSD) (HP: 0000078), cleft palate (HP: 0000175) and laryngomalacia. The psychomotor developmental milestones were reached late (HP: 0001263).

During her first two years of life, she underwent surgical correction of cardiac malformations, gastrostomy and tracheostomy. When she was two years old, she was hospitalized because of incoercible vomiting, and during this hospitalization she underwent ophthalmological consultations with evidence of a markedly pale optic disc at the fundus oculi (FO), normal cardiological examination, ultrasound of the complete abdomen with evidence of multiple mesenteric, globose, hypoechoic and diffuse mesenteric lymph nodes, presence of a fleeting and transient target image (“doughnut sign”) to be referred to ileal invagination in the left iliac fossa and resolved during the ultrasound examination.

At our evaluation, she presented microcephaly (0000252), a face with broad forehead, micrognathia, dysmorphic auricles with squared appearance and simplified design, thick eyebrows (0000574), synophrys (0000664), strabismus, short nasal bridge (0003196), dental diastema (0000175), slight thoracic fairing, telethelia, marked arachnodactyly with clenched fist, single palmar sulcus complete on the right and incomplete on the left hand. Di George syndrome and CHARGE were suspected and the patient underwent peripheral venous blood karyotype examination with FISH study of the 22q11.2 region, and *CDH7* analysis was executed. The karyotype was female normal (46, XX), the FISH study of the 22q11.2 region resulted in a normal hybridization pattern and *CDH7* analysis did not reveal any mutation.

### 3.2. Exome Sequencing and Variant Annotation

Clinical exome analysis revealed the *de novo* heterozygous frameshift deletion c.1722_1723delTG, p.Gly575SerfsTer2 in *RAD21* gene (NM_006265.2), which is associated with CdLS type 4 (OMIM #614701). We also searched for the same variant in the older sister, which was not found.

The variant p.Gly575SerfsTer2, confirmed by Sanger analysis (Figure 3), was not previously described in scientific literature, nor it was annotated in gnomAD or 1000G (Table 2), but was predicted to be highly pathogenic by CADD-based prediction. Moreover, although leading to a premature stop codon formation, it was predicted as Nonsense Mediated Decay (NMD) escaping (Table 2), forming a truncated protein lacking its C-terminal moiety. Finally, according to ACMG guidelines, it was classified as likely pathogenic (PVS1, PM2).

### 3.3. In Silico Prediction of RAD21 p.Gly575SerfsTer2

We performed protein-level analyses to assess the impact of the p.Gly575SerfsTer2 variant on protein function and structure. As shown in Figure 4a and stated in Table 2, this mutation occurs in the last exon (14/14) of the *RAD21* gene and was thus predicted to cause nonsense-mediated decay. Thus, we were prompted to compare wild type and truncated Rad21 three-dimensional structures using Phyre2 [18] and Protein Data Bank [17] (PDB) web tools. We found that the mutation caused a truncated protein missing the C-terminal secondary structures in the Smc1A binding domain, which includes several aminoacids required for the interaction with Smc1A itself and the stabilization of the whole cohesin-NIPBL-DNA complex (Cys585, Arg586, Gln592 and Leu603, among others) [13] (Figure 4b,c).

## 4. Discussion

Herein, we report clinical and genetic findings of a female patient with CdLS type 4, a syndrome of which the clinical features of only 30 patients have been previously described in the literature [13] (Table 3). CdLS type 4 individuals have long and smooth philtrum (HP: 0000319) (90%), short nose (HP: 0003196) (88%), concave nasal ridge (HP: 0011120) (83%), thick eyebrows (HP: 0000574) (83%), short fifth finger (HP: 0009237) (82%) and thin upper lip vermillon (HP: 0000219) (80%). The index patient presented with a clinical score of 11 pt (classic phenotype of CdLS) including the following features: short nose, thick eyebrow, global development delay, synophrys, thick eyebrows, microcephaly, weight < 2DS, height < 2DS. The most frequent malformations described in CdlS type 4 patients are brain imaging abnormalities (HP: 0410263) (50%), congenital heart defects (HP: 0000078) (39%) and cleft palate (HP: 0000175) (24%); our index patient presented cardiac anomalies (patent ductus arteriosus, PDA, and ventricular septal defect, VSD) cleft palate, dental diastema (HP: 0000699) which was found in only 2/20 patients (10%), gastrointestinal defects (HP: 0002012) found in 4/30 patients (13%), laryngomalacia and a markedly pale optic disc (HP: 0000543) at the fundus oculi which were never previously described in the literature. These percentages and the clinical data of our patient show that malformations are very frequent in CdLS type 4 and must be investigated further. Moreover, prenatal growth retardation was never previously observed in CdLS type 4.

The trio exome analysis showed the c.1722_1723delTG variant in the *RAD21* gene occurred *de novo* and was absent in the older sister. *RAD21* spans 29 Kb and has 14 exons, and they encode a protein of 631 amino acids. *RAD21* variants are found in a minority of CdLS patients: currently, only 10 missense variants and five microdeletions have been reported in CdLS patients [2,14].

In summary, we present the novel *RAD21* variant c.1722_1723delTG, p.Gly575SerfsTer2, giving rise to CdLS type 4 in a girl. Segregation data, bioinformatic analysis and population data indicate the pathogenicity of the novel variant. In silico structural modelling suggests that such pathogenicity is provided by the lack of the C-terminal moiety of the protein, which is detrimental to Smc1A binding and in turn for stability of the whole complex [13].

This report summarizes previously reported clinical manifestations of CdLS type 4, but also highlights new clinical symptoms, which will aid in correct counseling of future CdLS type 4 patients and appropriate follow-up.

## Figures and Tables

**Figure 1 genes-14-00119-f001:**
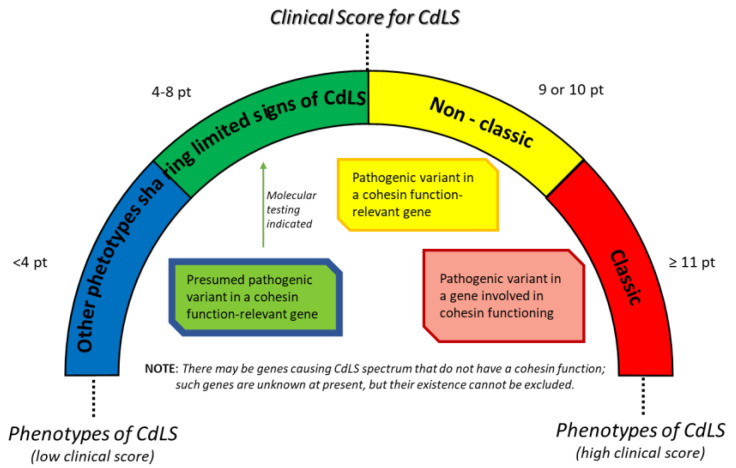
The rainbow phenotypes of Cornelia de Lange Syndrome (CdLS) [2].

**Figure 2 genes-14-00119-f002:**
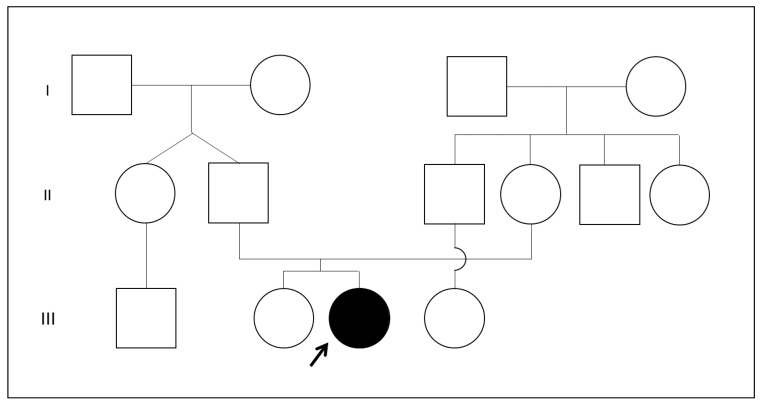
Family pedigree of our patient.

**Figure 3 genes-14-00119-f003:**
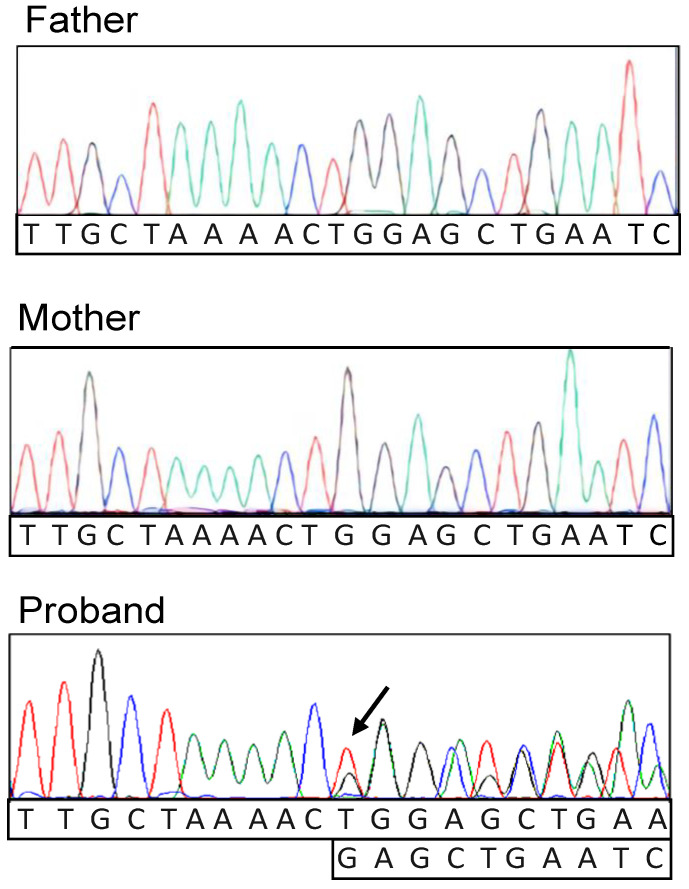
Trio sanger analysis showing the genotype of the parents and the proband at *RAD21* locus. The arrow indicates the NM_006265.2:c.1722-1723del in heterozygosis.

**Figure 4 genes-14-00119-f004:**
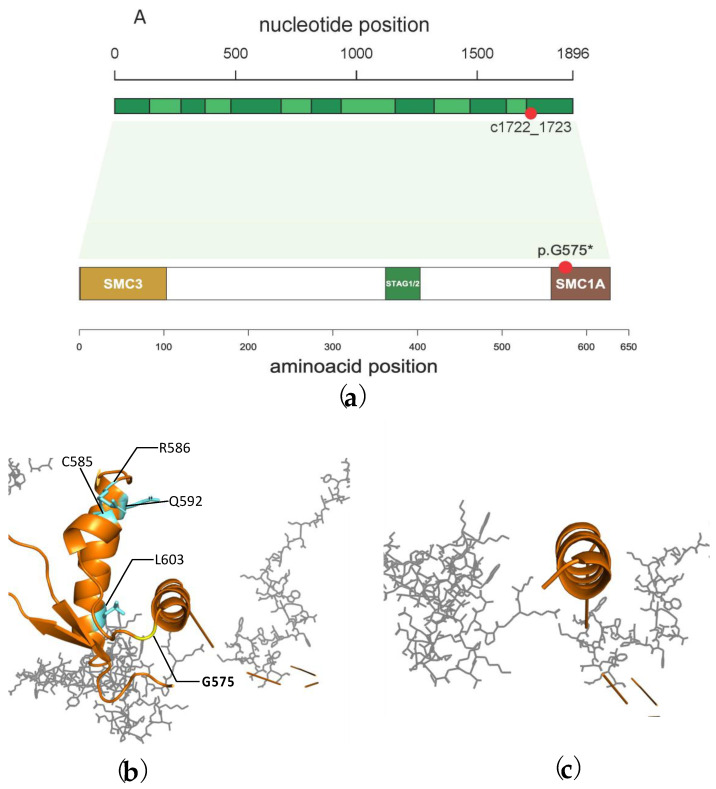
In silico prediction of *RAD21* p.Gly575SerfsTer2. (**a**) The frameshift variant occurs in the last exon of the coding sequence (Consensus Coding Sequence ID: CCDS6321.1, top), and is predicted to form a stop codon two residues downstream, in the SMC1A domain (bottom). (**b**) C-terminal moiety (last 80 residues, orange) of Rad21 wild type (**b**) or truncated (**c**) at position 577. In cyan, some residues important for Smc1A interaction [13]. The residue involved in the frameshift (Gly575) is depicted in yellow.

**Table 1 genes-14-00119-t001:** Clinical features of Cornelia de Lange Syndrome (CdLS) [2].

Clinical Features	Cardinal Features (2 Points Each If Present)	Suggestive Features (1 Point Each If Present)
**Facies**	Synophrys (HP: 0000664) and/or thick eyebrows (HP: 0000574)	Microcephaly (prenatally and/or postnatally) (HP: 0000252)
	Short nose (HP: 0003196), concave nasal ridge (HP: 0011120) and/or upturned nasal tip (HP: 0000463)	
	Long (HP: 0000343) and/or smooth philtrum (HP: 0000319)	
	Thin upper vermilion (HP: 0000219) and/or downturned corners of mouth (HP: 0002714)	
**Neuropsychomotor development**		Global developmental delay (HP: 0001263) and/or intellectual disability (HP: 0001249)
**Growth**		Prenatal growth retardation (<2 SD) (HP: 0001511)
		Postnatal growth retardation (<2 SD) (HP: 0008897)
**Musculoskeletal system**	Hand oligodactyly (HP: 0001180) and/or adactyly (HP: 0009776)	Small hand (HP: 0200055) and/or feet (HP: 0001773)
		Short fifth finger (HP: 0009237)
**Other major systems**	Congenital diaphragmatic hernia (HP: 0000776)	Hirsutism (HP: 0001007)
**Clinical Score**		
•≥11 points, of which at least 3 are cardinal → classic CdLS		
•9 or 10 points, of which at least 2 are cardinal → non classic CdLS		
•4–8 points, of which at least 1 is cardinal→ molecular testing for CdLS indicated		
•<4 points → insufficient to indicate molecular testing for CdLS		

**Table 2 genes-14-00119-t002:** Annotation features of NM_006265.2: c.1722_1723del variants.

**DNA variant (HGVSc)**	NM_006265.2: c.1722_1723del
**Genomic location (hg19)**	chr8: 117859911-117859913
**Gene Symbol**	RAD21
**Exon**	14/14
**Protein variant**	NP_006256.1: p.Gly575SerfsTer2
**Nonsense Mediated Decay (NMD) prediction**	NMD_escaping_variant
**CADD score**	35
**Allele Frequency (gnomAD)**	N/A
**Allele Frequency (1000G)**	N/A

**Table 3 genes-14-00119-t003:** Clinical characteristics of affected individuals with *RAD21* mutations compared to our index patient (grey column) and recalculation of the new percentages of the features of CdLS type 4 (CdLS-4). “+” stands for clinical features of CdLS-4 present in our patients while “−” stands for clinical features of CdLS-4 not present in our patient.

		30 Patients with CdLS-4		Our Patient	New Percentages (31 Patients with CdLS-4)
		N pos/N Total	Percentage		
	Sex (male/female)	15/15	50/50	Female	48/52
**Cardinal Features**	Synophrys (HP: 0000664) and/or	19/28	68	+	69
	Thick eyebrows (HP: 0000574)	20/24	83	+	84
	Short nose (HP: 0003196) and/or	23/26	88	+	89
	Concave nasal ridge (HP: 0011120) and/or	24/29	83	−	83
	Upturned nasal tip (HP: 0000463)	19/27	70	−	70
	Long (HP: 0000343) and/or smooth philtrum (HP: 0000319)	27/30	90	−	90
	Thin upper lip vermilion (HP: 0000219) and/or	24/30	80	−	79
	Downturned corners of mouth (HP: 0002714)	16/27	59	−	59
	Hand oligodactyly (HP: 0001180) and/or adactyly (HP: 0009776)	0	0	−	0
	Congenital diaphragmatic hernia (HP: 0000776)	1/30	3	−	3
**Suggestive Features**	Global development delay * (HP: 0001263) and/or	12/16	75	+	76
	—Cognitive functioning: normal cognition	3/29	10	−	10
	—Cognitive functioning: mild disability	13/29	45	−	45
	—Cognitive functioning: moderate disability	4/29	14	−	14
	Development problems (#)	7/29	24	−	24
	Intellectual disability (HP: 0001249), severity unspecified	2/29	7	+	10
	Prenatal Growth retardation (<2 SD) (HP: 0001511)	0	0	+	3
	Postanatal growth retardation (<2 SD) (HP: 0008897)				
	—Weight < 2DS	3/26	12	+	15
	—Height < 2DS	10/27	37	+	39
	Microcephaly (HP: 0000252) (prenatally and/or postnatally)				
	—Head circumference < 2DS	16/28	57	+	59
	—Brachycephaly	8/19	42	−	42
	Small hands (HP: 0200055) and/or feet (HP: 0001773)	5/27	19	−	19
	Short fifth finger (HP: 0009237)	23/28	82	−	82
	Hirsutism (HP: 0001007)	10/26	38	−	38
**Malformations**	Gastrointestinal defects (HP: 0002012)	4/30	13	+	16
	Genital abnormalities (HP: 0000078)	1/20	5	−	5
	Congenital heart defects (HP: 0000078)	9/23	39	+	42
	Cleft palate (HP: 0000175)	6/25	24	+	27
	Dental diastema (HP: 0000699)	2/20	10	+	14
	Brain anomalies (MRI brain) (HP: 0410263)	3/6	50	−	50

* as delay on one or more milestone. # too young to reliably determine cognitive functioning.

## Data Availability

Not applicable.

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
