# Peer review of "A Novel Variant in RAD21 in Cornelia De Lange Syndrome Type 4: Case Report and Bioinformatic Analysis"

_genes, 2023, doi:10.3390/genes14010119_

Round 1
Reviewer 1 Report
A new mutation in RAD21 in a 3-year-old girl with Cornelia de Lange syndrome type 4 was detected by exome sequencing and the findings will help the clinician correctly identify symptoms of this diverse syndrome. However, more pathogenic evidence should be done and there are aspects that require revision.
1. Results: Despite this detailed description of the index patient's phenotype, it would be preferable to have some photos of the patient.
2. Results: The family's pedigree map is missing.
3. Results: The patient was the second of two children. Please include the first child's genotype and phenotype.
4. Figure 1: There is some misunderstanding about rainbow phenotypes. What is the significance of the color red? Except for the clinical score, does it have the same meaning as the yellow one?
5. Figure 2: Please identify the family members and the precise base sequence.
6. Figure 3(b、c): It's not very evident in the schematic three-dimensional representation, and it would be excellent to zoom in to show WT and del. (eg similar to figure 2 DOI: 10.1016/j.ejmg.2018.08.007 )
7. the mutation pathogenic should be functional verification.
Author Response
We thank the Referees for having carefully read our paper and for having provided some
useful comments. We have submitted a revised version of the manuscript entitled “A novel variant in RAD21 in Cornelia de Lange syndrome type 4: case report and bioinformatic analysis” (genes-2094653), considering the comments received by the two referees. Please find below the answers to the reviewers’ reports. We have marked via the revised command the modifications done in the paper.
A new mutation in RAD21 in a 3-year-old girl with Cornelia de Lange syndrome type 4 was detected by exome sequencing and the findings will help the clinician correctly identify symptoms of this diverse syndrome. However, more pathogenic evidence should be done and there are aspects that require revision.
- Results: Despite this detailed description of the index patient's phenotype, it would be preferable to have some photos of the patient.
Unfortunately, it was not possible to take pictures of the patient due to the lack of consent by the parents.
- Results: The family's pedigree map is missing.
We have now added the family’s pedigree map (see new Figure 2).
- 3. Results: The patient was the second of two children. Please include the first child's genotype and phenotype.
We added clinical information of the first child including genotype and phenotype (lines 101-103, 134-135 and 239).
- 4. Figure 1: There is some misunderstanding about rainbow phenotypes. What is the significance of the color red? Except for the clinical score, does it have the same meaning as the yellow one?
We changed the color of the figure 1 because it was misunderstanding and we have deeply described the rainbow phenotypes (lines 45-49).
- Figure 2: Please identify the family members and the precise base sequence.
We marked the family members and we show the precise base sequence.
- Figure 3(b、c): It's not very evident in the schematic three-dimensional representation, and it would be excellent to zoom in to show WT and del. (eg similar to figure 2 DOI: 10.1016/j.ejmg.2018.08.007 )
Taking as an example the figure 2 of the above cited paper, we properly zoomed the three-dimensional representation of Rad21 at C-terminal moiety, also depicting amino-acid which are involved in Smc1A interaction. Please note that former figure 3 now has turned into figure 4.
- the mutation pathogenic should be functional verification.
All the in silico predictions suggest the pathogenicity of the NM_006265.2:c.1722_1723del variant. Although functional analysis will be necessary to confirm the predicted pathogenicity, this goes beyond the aim of this paper.
Reviewer 2 Report
The manuscript expand the clinical description of CdLS type 4 by reporting the clinical features of a girl with a novel mutation in RAD21 and also summarizes previously reported clinical manifestations of CdLS type 4
Some aspects that need consideration:
1. I would suggest to the authors to include Human Phenotype Ontolgy (HPO) codes when they described patient clinically and also in Table 1
2. Has it been checked that it is a variant not reported in the general population? .Please, specify which databases were consulted.
Have you tried to get photos of the patient? This would make it possible to compare the dysmorphic features with other patients.
3. Do they have the approval of a local ethics committee for the studies and analyses of the patient?
4. The discussion should be deeper into the phenotype relationship with the other previously reported patients ( CdLS type 4)
5. Further discussion of structural modelling and RAD21 interaction with SMC1 is needed.
Author Response
We thank the Editor and Referees for having carefully read our paper and for having provided some
useful comments. We have submitted a revised version of the manuscript entitled “A novel variant in RAD21 in Cornelia de Lange syndrome type 4: case report and bioinformatic analysis” (genes-2094653), considering the comments received by the two referees. Please find below the answers to the reviewers’ reports. We have marked via the revised command the modifications done in the paper.
The manuscript expands the clinical description of CdLS type 4 by reporting the clinical features of a girl with a novel mutation in RAD21 and also summarizes previously reported clinical manifestations of CdLS type 4.
Some aspects that need consideration:
- I would suggest to the authors to include Human Phenotype Ontolgy (HPO) codes when they described patient clinically and also in Table 1
We have included the HPO codes in the text (in proband phenotype section and in the discussions) and in Table 3.
- Has it been checked that it is a variant not reported in the general population? .Please, specify which databases were consulted.
We searched for this variant in the main publicly available databases of allele frequency (gnomAD, ExAC and 1000G), but we did not find it in any population, as indicated in the new table 2.
- Have you tried to get photos of the patient? This would make it possible to compare the dysmorphic features with other patients.
Unfortunately it was not possible to take pictures of the patient due to the will of the parents
- Do they have the approval of a local ethics committee for the studies and analyses of the patient?
Our study has been approved by our local ethics committee (lines 79-81).
- The discussion should be deeper into the phenotype relationship with the other previously reported patients ( CdLS type 4).
In the Discussion section, we have better clarified the genotype-phenotype relationship in the patients affected by CdLS 4. We focused about malformations and the most relevant signs of the syndrome compared to the clinical features of our index patient. In order to better clarify this section we have bettered the Table 3. Please note that former Table 2 now has turned into Table 3.
- Further discussion of structural modelling and RAD21 interaction with SMC1 is needed.
We provided further information on structural modeling and Rad21-Smc1A interaction in Methods and in Discussion Section, respectively (lines 95-98 and 245-248).
Round 2
Reviewer 1 Report
all questions are revised except the function identified.